# Epidemiology of Chronic Hepatitis C in First Nations Populations in Canadian Prairie Provinces

**DOI:** 10.3390/pathogens14070693

**Published:** 2025-07-14

**Authors:** Kate P. R. Dunn, Dennis Wardman, Maxim Trubnikov, Chris Sarin, Tom Wong, Hongqun Liu, Samuel S. Lee

**Affiliations:** 1Faculty of Health, York University, Toronto, ON M3J 1P3, Canada; katedunn@yorku.ca; 2Clinical Addiction Medicine Practice, Regina, SK S4T 0X4, Canada; dwardman@shaw.ca; 3Indigenous Services Canada, Ottawa, ON K1A 0K9, Canada; maxim.trubnikov@sac-isc.gc.ca (M.T.);; 4Indigenous Services Canada, Edmonton, AB T5J 4G2, Canada; chris.sarin@sac-isc.gc.ca; 5School of Epidemiology & Public Health, University of Ottawa, Ottawa, ON K1S 5S9, Canada; 6Dalla Lana School of Public Health, University of Toronto, Toronto, ON M5S 1A1, Canada; 7Liver Unit, University of Calgary Cumming School of Medicine, 3330 Hospital Dr NW, Calgary, AB T2N 4N1, Canada; hliu@ucalgary.ca

**Keywords:** hepatitis C, First Nations health, Indigenous, epidemiology, colonial trauma, prevalence, incidence, racism

## Abstract

Current structural barriers experienced by First Nations in Canada shape access and engagement for testing and treatment of hepatitis C virus (HCV) infections. This non-systematic informative review considers transdisciplinary perspectives, regional data, and published literature connecting context to the disproportionate HCV burden experienced by First Nations populations in the prairie provinces of Canada, and offers examples of participatory and community-led initiatives working toward the elimination of HCV as a public health threat. First Nations in Canada are disproportionately impacted by chronic HCV infection, with a reported rate of newly diagnosed HCV cases in First Nations communities five times the respective rate in the general Canadian population in 2022. This review explores the reasons underlying the disproportionate burden of HCV infection. Significant over-representation of First Nations in the Canadian Prairies is seen in the major risk categories for HCV acquisition, and the impact of these risk factors is aggravated by barriers to accessing healthcare services and medication coverage. These barriers stem from the legacy of colonialism, discrimination, disenfranchisement, and are exacerbated by stigmatization, victimization, and racism in the justice and healthcare systems. Other contributory factors that impede access to care include inadequate healthcare clinic staffing and infrastructure in First Nations communities, and significant geographical distances between First Nations reserves and laboratories, pharmacies, and treating/prescribing healthcare providers. Recent recognition of historical harms and early steps towards nation-to-nation reconciliation, along with support for culturally connected, holistic, and First Nations-led wellness programs, instill hope that elimination strategies to eradicate HCV infection in First Nations populations will be successful in Canada.

## 1. Introduction

Indigenous populations in what is now called Canada consist of not only the three distinct groups known as First Nations, Inuit and Métis, but encompasses over 70 languages, unique origin stories, more than 600 Nations, over 3400 reserved land locations, and over 1.8 million people, constituting 5% of the population in Canada [1]. According to the 2021 census, self-identified Inuit, Métis, and First Nation persons constituted 0.2%, 1.7%, and 2.8%, respectively, of the entire Canadian population. Indigenous Peoples are not only the fastest growing population segment at >9% growth but also the youngest population in Canada, where over a quarter are younger than 25 [1]. This description gives the impression of vibrant sovereign Peoples thriving within one of the largest and healthiest countries of the world, but the lived reality retains shadows of history routinely experienced in systemic and structural barriers shaping significantly disparate life expectancy rates and blatant access hurdles to equitable healthcare.

The disparate experience of HCV by First Nations provides a clear health problem calling for reflective collaboration and analysis of published literature in connection with current data to inform healthcare solutions. The nuances of maneuvering aspects of culture, science, quantitative data, qualitative perspectives, community-based participatory action research, and gaps in the available literature necessitated a non-systematic informative review approach combining diverse strategies and a transdisciplinary team of practitioner authors for this article. Through this collaborative synthesis approach, the authors seek to increase awareness of the value of context and history in shaping impactful infectious disease strategies.

An in-depth understanding of the factors underlying the high rates of HCV in Indigenous People in Canada, especially for non-Canadian readers, requires an explanation of terms (see Appendix A—Glossary (includes References [2,3]), as well as understanding the role and impacts of the colonial legacy of HCV and institutional racism in Canadian healthcare. These are provided in Appendix A (includes References [3,4,5,6,7,8,9,10,11,12,13,14,15,16,17,18,19])

## 2. Methodology

The authors comprise a multidisciplinary group with long experience and expertise in communicable disease epidemiology, public health, infectious diseases, nursing, hepatology, and network databases, especially pertaining to Indigenous Peoples. This is not a systematic review. Although Indigenous Peoples are disproportionately impacted by HCV, there is a gap in the literature showing inclusion of Indigenous perspectives in HCV initiatives, funding, and programming. A Google Scholar search using the terms ‘Indigenous hepatitis C Canada’ produced approximately 33,200 publications, and a review of the first 200 highlights this gap. Most articles briefly mention Indigenous Peoples as a priority population, many mention data supporting the increased prevalence of HCV in Indigenous Peoples without actually showing the detailed data, several link the current experience of high HCV rates as a colonial legacy, while a few illustrate models of care, connection to culture, or wellness-based initiatives seeking healing or redirection of the current disparaging storylines. We attempted to select those papers on First Nations HCV epidemiology and care pathways that were directly relevant to the themes of this review or showed original data. We made no attempt at a systematic or comprehensive review given the vast number of incompletely documented or incompletely collected papers in this field, especially in the so-called ‘grey literature’ of non-peer-reviewed abstracts and conference proceedings. A PubMed search using the terms ‘First Nation’, ‘hepatitis C’, and ‘Canada’ yielded 138 papers. These were reviewed by the authors and included if they contained relevant data or observations.

This review focuses on First Nations populations in the three Prairie provinces, Alberta, Saskatchewan, and Manitoba (Appendix A), because HCV epidemiologic data in Métis and Inuit, especially in other regions of Canada, are incompletely or poorly collected/collated.

Original epidemiologic data provided by Indigenous Services Canada (ISC) are reported herein with the permission of the ISC Public Health program.

### 2.1. Hepatitis C Virus Infections in First Nations: A Historical Perspective on the Data Systems and Reported Outcomes

HCV is a blood-borne virus whose chronic infection wreaks havoc on liver function, results in more life years lost than any other infectious disease in Canada, and creates considerable risk for further liver disease and hepatocellular cancer [20]. Canadian National Health Surveys are generally not inclusive of the First Nations populations residing on-reserve, Indigenous Peoples within prison populations, and under-representative of Indigenous Peoples residing in northern, remote, and isolated communities [21]. This results in national health and social data collection instruments (e.g., Census, national health surveys, etc.) continuously missing the mark with both enumeration of (“the denominator problem”) and collecting outcome data from (“the numerator problem”) First Nations, Métis, and Inuit.

There is a paucity of HCV surveillance and clinical outcomes data in Indigenous populations in Canada, especially in Métis and Inuit populations [21,22]. In First Nations, HCV surveillance data availability is relatively “better”, though not as comprehensive or timely as it is in the general Canadian population.

A recent comprehensive environmental scan of grey literature and peer-reviewed projects, programs, and initiatives in the province of Saskatchewan published between 1995 and 2019 yielded only three HCV-specific academic research results, inclusive of both original research and reviews [23].

Available surveillance data and peer-reviewed publications describing regional perspectives suggest HCV burden in First Nations communities in Canada has increased over the past 30 years. Between 1993 and 2002, the cumulative positive antibody or anti-HCV incidence rate in First Nations in Manitoba was 91.1 per 100,000, or 2.5 times the respective rate in the non-First Nations individuals in Manitoba at 36.6 per 100,000 [22,24,25]. Between 2010 and 2015, Gordon and colleagues reported anti-HCV incidence rates in First Nations individuals residing in First Nations communities in Northern Ontario ranging from 56.6 per 100,000 in 2010 to 324.2 per 100,000 in 2015 [26]. In 2015, the anti-HCV incidence rate in First Nations in Northern Ontario was 11 times the rate within the general Ontario population for that year [26]. Younger adults aged 20–29 years made up 45%, while women made up 52% of the HCV antibody-positive population during the study period [26]. The geographical variability in HCV epidemiology likely reflects the large number of often diverse Indigenous cultures and communities with differing levels of socioeconomic status, proximity to larger centers and healthcare access, traditions, healthcare practices, and beliefs.

In 2019, the Public Health Agency of Canada (PHAC) estimated anti-HCV prevalence in First Nations in Canada at 7.4% with a plausibility range of 3.5% to 11.2% of the total First Nations population [21]. Approximately 3.5% with a plausibility range of 2.0% to 5.0% of the total First Nations population in Canada were estimated to live with chronic hepatitis C infection and require treatment. These rates were more than seven times the estimated anti-HCV prevalence and HCV RNA rates in the general Canadian population [21]. It is of note that the upper bound of this estimate was produced by PHAC in partnership with First Nations Health Services Organizations in Alberta and Saskatchewan and Indigenous Services Canada and released as a part of results of a bio-behavioral survey among First Nations populations in 2018–2020 in the provinces of Alberta and Saskatchewan [27].

### 2.2. HCV Surveillance in First Nations in Canadian Prairies

Case definitions of HCV incidence vary slightly between the 3 provinces. These are detailed in Appendix A, along with a discussion of the limitations of these methodologies.

Between 2011 and 2023, the (crude) reported rate of newly diagnosed cases of HCV infection in First Nations communities in the Prairies provinces of Alberta (AB), Saskatchewan (SK) and Manitoba (MB) almost doubled from 78.5 per 100,000 in 2012 to peaking at 140.3 per 100,000 in 2018 (Figure 1). By 2023, the reported rate (82.0 per 100,000) was 42% lower than that in 2018 (Indigenous Services Canada, unpublished data, 2024) (Figure 1).

During the same time, the rate of newly reported cases of HCV infection in the general Canadian population increased only by 15% from 29.5 per 100,000 in 2011 to peaking at 33.9 per 100,000 in 2018 (Figure 1). By 2022 (the most recent year with available national statistics), the reported Canadian rate of 18.3 per 100,000 was 46% lower than that in 2018 [28]. The ratio of the reported rates for newly diagnosed HCV infection between First Nations living on-reserve and the general Canadian population has widened, from 2.7 in 2011 to 5.1 in 2022 (Figure 1).

While rates of HCV diagnoses in First Nations males (81.2 per 100,000) and First Nations females (82.8 per 100,000) residing in AB, SK, and MB communities were similar in 2023, when stratified by age group and sex, the highest age-specific rate was in younger females and older males, illustrating the need for tailored age- and sex-specific public health interventions (Figure 2), and highlighting the interconnected issues around power dynamics, partner violence, and economic dependence.

For comparison, the reported incidence rate of HCV among males in the general population in Canada for 2020 was 22.9 per 100,000, and the rate among females was 13.7 per 100,000 [29].

### 2.3. Anti-HCV Status Awareness and Linkage to Care Among First Nations

While estimates of awareness of HCV status in First Nations populations are lacking, a recent national analysis suggested 24% of Canadians with evidence of past or current HCV infection were not aware of their infection status [21], while earlier analyses put it between 44% [30] and 60% [31]. In 2022, the reported linkage to HCV care among HCV RNA-positive First Nations populations in Alberta and Saskatchewan was approximately 60% [27].

### 2.4. Sources of HCV Surveillance Data and Data Limitations

As a notifiable disease, all newly diagnosed HCV infections among reporting First Nations communities between 2011 and 2023 were reported to the public health departments. Infections reported between 2020 and 2022 should be interpreted with caution due to the impact of reallocation of health resources in response to the COVID-19 pandemic, resulting in reduced testing for other communicable diseases, including HCV. Aggregate incidence rates were calculated using both case and population counts provided by ISC regional surveillance teams. Data for all three Canadian regions (MB, SK, and AB) were included for all reporting years (2011–2023).

### 2.5. The Colonial Legacy Link and Transmission of HCV

A devastating effect of the colonial legacy in Canada is the disproportionately high prevalence of HCV among First Nations. The simplistic explanation is that First Nations are significantly and disproportionately overrepresented in all the significant behavioral and social determinants or factors that lead to HCV acquisition, including substance use, incarceration, homelessness or crowded housing, non-sterile tattoo or piercing, and other determinants of risk activity for HCV acquisition and transmission. Of these factors, the most extensive dataset can be obtained by examining rates of federal and provincial incarceration, where Indigenous (First Nations, Inuit, and Métis) Peoples are sevenfold more likely than non-Indigenous people to be incarcerated in a federal institution [32,33]. In the provincial prisons included in ten provinces, the range varies from approximately 2-fold greater incarceration rate in Nova Scotia all the way to a shocking 23-fold in the province of Saskatchewan [10] (Over-representation of Indigenous persons in adult provincial custody, 2019/2020 and 2020/2021, Table 1).

Beyond the simplistic explanations, examining the root causes of these high incarceration rates shows a disturbing picture of disenfranchisement, discrimination, racism in the justice system, and rupture of traditional social and family supports. Muir et al. [34] studied a cohort of Indigenous (First Nations, Inuit, and Métis) adults in three Ontario cities, ranging from smaller, medium-sized, and large (Thunder Bay, London, and Toronto, respectively). They found that three factors contributed to high rates of incarceration: experiences of racism, removal from and/or disturbance of family support, and victimization [34].

Compounding the overrepresentation in determinants of HCV acquisition noted above, other factors sharply exacerbate ongoing HCV transmission: high rates of food insecurity and resulting malnutrition, inadequate access to basic healthcare, lack of HCV awareness and testing, reduced access to curative therapy, coexisting with ongoing vectors for continuing horizontal transmission. In their comprehensive review, Fayed and colleagues (2018) exposed colonialism as a “direct determinant of the inequity in HCV burden among Indigenous people…” and provided a pathway for decolonization of HCV care in Canada [35]. Both the existence of the gaps in HCV burden in First Nations and in the availability of health outcomes data on HCV in Indigenous populations in Canada were attributed to the colonial mindset that has been guiding both HCV research and the response to the HCV epidemic in Canada to date [35].

### 2.6. Access to Hepatitis C Testing and Treatment

Direct-acting antiviral (DAA) treatment options are available at no cost to eligible Canadians, including First Nations, providing an improved treatment experience for HCV. Although sustained virologic response (SVR) rates after treatment completion in First Nations match those of non-First Nations populations, the rates of engaging in and completing treatment are markedly lower [36,37,38]. Because access to prevention, harm reduction, testing, and care varies widely among jurisdictions, the prevalence of HCV varies as well and is reported to be from four to eleven times that of the general population. Variability is evident between First Nations members living off reserve in Ontario, Canada, with longer median time from a positive HCV antibody test result to HCV-RNA testing than individuals living on reserve (288 days off reserve versus 68 days on reserve) [39]. It is notable that 17% with a positive HCV antibody do not follow up with the next step to complete the HCV-RNA testing, and of those who do, 60% do not initiate treatment [40].

The Canadian provinces of Alberta (Figure 3) and Saskatchewan (Figure 4) have limited on-reserve First Nations data. In Alberta, the highest rates of HCV occur in off-reserve First Nations People (Figure 3). The COVID-19 pandemic may have decreased testing and access to testing in recent years, which could explain the recent decrease in HCV incidence. Efforts are underway to increase awareness and access to testing using standard serological testing as well as dried blood spot (fingerprick blood droplets placed on a card and mailed for laboratory HCV antibody and RNA processing) and point-of-care rapid testing.

Geographical factors also contribute significantly to the access to care barrier in First Nations in the Prairie provinces. Most First Nations are small, isolated, and remote communities far from large urban centers, and their healthcare clinics suffer from poor infrastructure, facilities, and staffing. Thus, First Nations persons must often travel significant distances to access laboratories, testing, pharmacies, or prescribing physicians or nurse-practitioners. Appendix A shows all the small First Nations reserves in the three Prairie provinces, which have a large land area about the same size as France, Spain, Germany, and Italy combined. In Alberta, for example, only 18% of the 53 First Nation communities have on-site access to labs, while 36% must drive <30 min to a lab, and 46% must drive between 30 min to 3 h to access lab testing (Dunn KP, unpublished observations, 2022).

Access to specialist physicians who can prescribe curative direct-acting antiviral (DAA) therapy is also hindered by geographical distance. Although in Alberta, all doctors, nurse-practitioners (NP), and even pharmacists can prescribe DAAs, it is the only province in Canada that permits this, and in the other provinces and territories, only specialist physicians licensed in gastroenterology, hepatology, infectious disease, or addiction medicine can prescribe DAAs. Thus, in the other regions of Canada, the Canadian healthcare model of so-called “specialist-driven” medicine where management of conditions such as HCV is carried out by specialists who only see patients after referral from a generalist physician or nurse-practitioner, leaves many Indigenous patients unable to access curative therapy because their local reserve clinic often has inadequate or only sporadic staffing/visits by physicians and NPs.

### 2.7. Colonial Influence on the Experience of Hepatitis

The etiology of the current disparate experience of hepatitis by First Nations populations in Canada cannot be discussed without clear linkage to colonial disruptions that continue to the present time. The Indian Act of 1876, although enacted in the last century with the intent to terminate Indigenous culture and the social, political, and economic relationships shaping distinct Indigenous Peoples, continues to dictate the funding and governance structure for health, education, lands, and leadership. Indigenous Peoples living through, or having family members who experienced the colonizing disruptions and confines of forced attendance in residential school, being removed from their family by the child welfare system, or experiencing the injustices of the judicial system where we see a disparate number of Indigenous Peoples incarcerated, continues to impact not only current physical, mental, emotional and spiritual well-being but also the disruption of interpersonal relationships in the next generations, increased rates of numerous infectious and chronic diseases, and the draw to addictions as coping mechanism as seen through multiple generations [35,42].

Research with Indigenous young people conducted in British Columbia showed a significant association between childhood maltreatment (residential school, child welfare system, foster care, and sexual abuse) and HCV infection [16]. Many of the socioeconomic systems in Canada conceived at the time of the Indian Act still influence healthcare services development and access to them, as evidenced by the reticence of Indigenous communities and community members to engage in HCV treatment [38,39,40].

### 2.8. Progress Toward HCV Elimination by 2030 in First Nations

Efforts toward the World Health Organization (WHO) goal of hepatitis elimination as a public health threat by 2030 outlined in the *Blueprint to Inform Hepatitis Elimination Efforts* identify priority populations as: People who are incarcerated (PWI); People who inject drugs (PWID); Indigenous Peoples (First Nations, Inuit, and Métis); Gay, Bisexual and other men who have sex with men; Newcomer and Immigrants from countries with high prevalence rates; and people born between 1945 and 1975 [43]. Over-representation of Indigenous Peoples (First Nations, Inuit, and Métis) is evident in several of these priority populations, skewing not only the data but also the stigma [44,45]. Among PWUD, we must consider the sociodemographics within 28% of that group who also identify as Indigenous [45]. The dual or intersectional stigma of racism, and HCV status, or identification with more than one of the priority population groups, not only complicates the personal experience of interactions with healthcare systems but often compounds self-stigma, which may be seen in non-commitment to testing, treatment, or as loss to follow-up [40,44,45,46]. By identifying Indigenous populations as a priority group, are we actually supporting health equity or further stigmatizing an already heavily stigmatized population group?

These problems of stigmatization, combined with a lack of community awareness, the asymptomatic nature of chronic HCV infection, poor healthcare facilities and staffing, lack of funding, and geographical remoteness, have severely curtailed any real prospect of HCV elimination by 2030 in First Nations.

Recent modelling analysis [47] indicates that most Canadian provinces except Manitoba, Ontario, and Quebec are on track to meet the WHO 2030 target, but the specific subpopulations of First Nations persons in these provinces was not addressed because data were lacking, although those authors commented that Indigenous populations across Canada represent a subgroup that is very unlikely to achieve the elimination target by 2030 [47]. In part, this is because the above factors dramatically reduce HCV testing, and thus an unknown percentage, but likely a large percentage, if not the majority, of Indigenous HCV-infected persons are unaware of their infection. As with all other populations, the COVID pandemic severely reduced the number of cases tested and detected [47].

### 2.9. Strategies for Change Through Promising Practice

In HCV care, there continues to be gaps in equity, access, support, and culturally competent care. However, especially over the past decade or so, significant improvements have started, or are in progress.

First Nations patients may be more comfortable accessing care with Indigenous healthcare providers; thus, medical and nursing schools have undertaken significant measures to increase enrollment of Indigenous students. These programs have been developed with the cooperation of the Indigenous Physicians Association, Indigenous Nurses Association, Office of Indigenous Health at the Royal College of Physicians and Surgeons of Canada, the Indigenous Health Committee at the College of Family Physicians, and Indigenous Services Canada.

Other measures include telehealth options, supporting remote access to HCV diagnosis and therapy, mobile outreach services, dried blood spot testing, rapid point-of-care testing for antibody, viral load, as well as innovations such as at-home testing, and reducing current bureaucratic hurdles to lab diagnosis and treatment payment/reimbursement.

Creating a supportive environment for substance use disorder recovery, or a recovery capital approach, can be a useful model, which by definition includes internal and external resources to initiate and sustain recovery. Such approaches go beyond simple clinical approaches, and include family and community capital in shaping a healing environment to support long-term success [46,48].

There are several examples across Canada of connected care models. The Alberta ECHO HCV program allows virtual access to a prescribing hepatologist for liver disease consults and HCV treatment in remote First Nations and Métis communities [49,50]. Through care conversations with Indigenous healthcare providers in Alberta, the need for relevant HCV awareness resources was raised, and First Nations Knowledge Keepers collaborated to co-create and produce a DocuStory film showcasing HCV awareness through a liver wellness narrative approach [51].

Dried blood spot testing has been implemented as an option offering destigmatizing and low-barrier testing for multiple health concerns, including HCV antibody and HCV RNA through a finger prick [52]. This creates testing options for home use, health fairs, mass screening, anywhere a lab is not readily accessible, and is an option for those who do not access mainstream healthcare facilities, while providing an opportunity to build relationships, reconnect for results, and support wellness.

Nurse-led programs across Canada support high levels of HCV testing and treatment success in Indigenous communities [53]. Saskatchewan has co-created multidisciplinary community-led models focusing on de-stigmatization and increasing access to HCV treatment utilizing a mentored connection [54]. British Columbia co-created a ‘seek and treat’ model for micro-elimination working through nurses who visited people and their friends or family connections in their supportive housing sites to provide HCV testing and treatment [55].

Addiction, medicine, and urban supportive care centers also include HCV awareness, prevention, testing, and treatment alongside social supports and harm reduction approaches to provide wrap-around supports in many centers, including Toronto [53], Vancouver [55], and Calgary [56].

New initiatives provide testing, treatment, or linkage to care for incarcerated people [57]. Although these programs may not be Indigenous-led, they provide wellness supports for both Indigenous and non-Indigenous inmates.

There are several Indigenous-led efforts supporting destigmatization and holistic perspectives across Canada. An example is the work of Kimamow Atoskanow Foundation in Alberta [58], shaping rural and land-based supports around wellness and cultural teachings on sexual health. Communities, Alliances and Networks (CAAN) is an Indigenous-led organization partnering with funders and national stakeholders to support HCV and HIV community readiness resources, education, funding, and programming [59].

The initiatives mentioned above are not a comprehensive list; instead, these examples are mentioned to encourage Indigenous-led advocacy in impacting current policy, inadequate data availability, and initiating or supporting community-based programming.

## 3. Limitations

There are several limitations to this review. First, as acknowledged in the Methodology section, data in the Inuit and Métis populations are significantly lacking or incompletely collected; thus, our original intent to review the entire Indigenous HCV epidemiology was not feasible. Accordingly, this review only focuses on First Nations. However, this underscores the pressing need to collect accurate data on the epidemiology and management of HCV in Inuit and Métis in Canada. Moreover, there are significant gaps even in First Nations data. Specifically, as detailed in the above sections on the epidemiology of HCV in each Prairie province, a number of knowledge gaps remain. A persistent knowledge gap is the exact geographical distribution of First Nations Peoples with HCV, specifically whether they live on reserve or off reserve. Many papers in the literature are also perhaps slightly inaccurate because First Nations status was not verified through the Indian Registration system due to privacy concerns, but was instead through self-identification. Again, as mentioned previously, First Nations data in other regions of Canada are less completely collected, and thus, we limited this review to the three Prairie provinces. The three provinces of Alberta, Saskatchewan, and Manitoba contain the largest proportions of self-reported Indigenous Peoples, at 7%, 17%, and 18% of the total populations of each province, according to the 2021 Canadian census [1]. Overall, in Canada, 5% of the population self-identifies as Indigenous [1]. Thus, our review in these regions, where Indigenous Peoples comprise a relatively larger segment of the population, may not be directly applicable to provinces such as Ontario, where they comprise only 2.5% of the population [1].

Finally, the literature shows a significant gap in that much of it is written by non-Indigenous authors. There is a significant paucity of HCV studies led by Indigenous authors representing First Nations perspectives on issues such as epidemiology, care pathways, destigmatization, barriers to treatment, culturally relevant engagement practices, etc. Moreover, few studies include the perspectives of First Nations persons with lived experience of HCV. We hope that future studies will take note of these gaps and attempt to address them.

## 4. Conclusions

The high rates of HCV prevalence and annual incidence in First Nations in Canada are a legacy of the colonial trauma of the past that continues in many respects to the present day. First Nations are overrepresented in most of the behavioral determinants that lead to HCV acquisition. These include substance use, incarceration, homelessness, and substantial barriers to culturally safe prevention, high-quality testing, curative DAA therapy, and care. As the data indicate, there are regional variations, so focused strategies are needed. Furthermore, there are specific needs to provide interventions for young Indigenous females, who appear to be disproportionately impacted. Many opportunities exist for improving the relationship between the justice system, police services, corrections infrastructure, and First Nations to address the racism, inequity, and abuses both in judicial policy, as well as improving healthcare access and HCV treatment access within these structures.

In recent years, there has been widespread acknowledgment by non-Indigenous people and levels of government that Indigenous Peoples have suffered and continue to suffer from racism, discrimination, and disenfranchisement. Along with this, there are now many proclamations, projects, and plans to start addressing these inequities in a meaningful manner.

The COVID-19 pandemic demonstrated that Indigenous communities can implement focused in-community testing for infectious diseases, and localized approaches are effective at educating community members on health topics as well as conducting screening and providing personal and supportive holistic care. It is also striking how factors affecting initial engagement in care and retention in supportive care are the pivotal piece in contributing to achieving SVR. We are hopeful that these important measures will eventually result in the elimination of HCV in First Nations in Canada.

## Figures and Tables

**Figure 1 pathogens-14-00693-f001:**
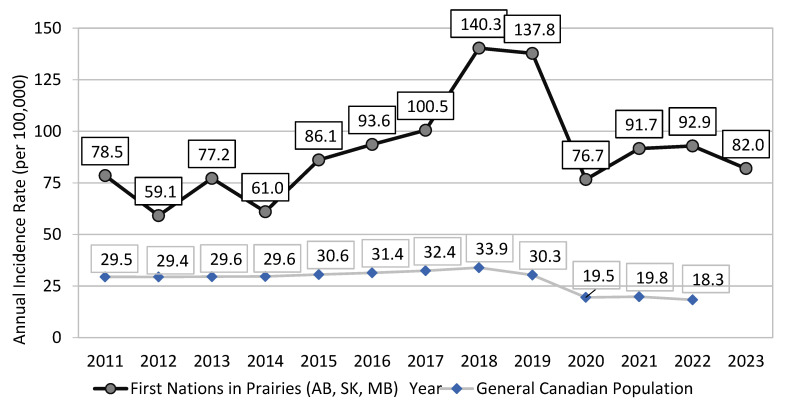
Rate (per 100,000) of newly reported cases of HCV among First Nations communities in AB, SK, and MB vs. the general Canadian population, 2011–2023. (Source: Indigenous Services Canada, 2024).

**Figure 2 pathogens-14-00693-f002:**
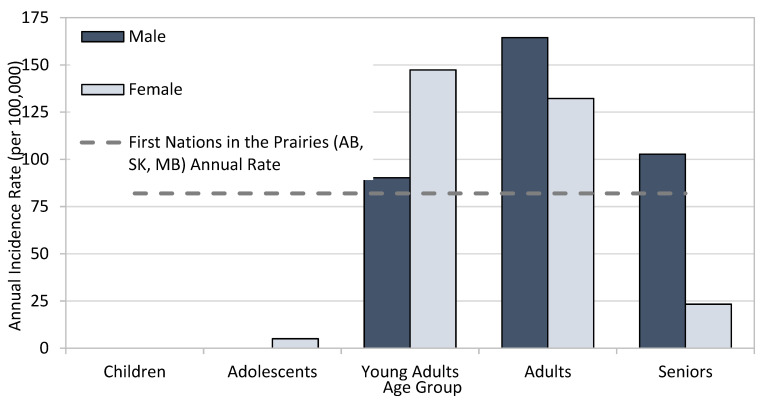
Annual rate (per 100,000) of newly reported cases of HCV among First Nations living in AB, SK, and MB communities by age group and sex, 2023 (Source: Indigenous Services Canada, 2024).

**Figure 3 pathogens-14-00693-f003:**
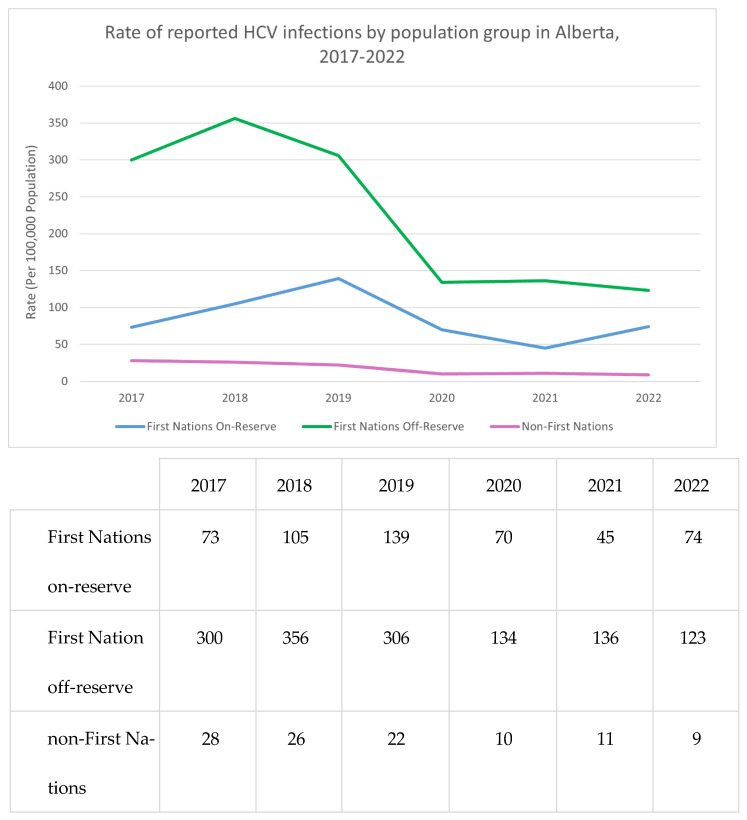
Rate of reported HCV infections by population group in Alberta, 2017–2022. Sources: FNIHB AB CDC Database; Government of Alberta, Alberta Health; ISC, Indian Registry.

**Figure 4 pathogens-14-00693-f004:**
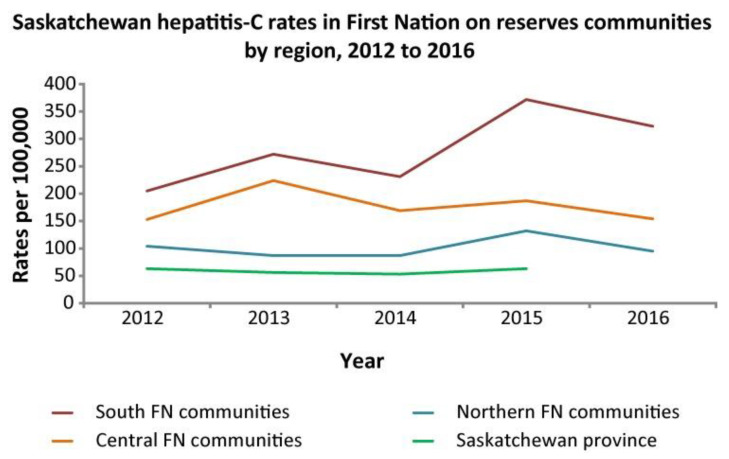
HCV diagnosis rates in Saskatchewan province per 100,000 population, 2012–2016. Reproduced from Skinner et al. [41].

**Table 1 pathogens-14-00693-t001:** Comparison of incarceration rates between Indigenous and non-Indigenous populations in selected provinces, 2020/2021 data *.

Province	Non-Indigenous (per 10,000)	Indigenous (per 10,000)
Saskatchewan	4.3	100.7
Alberta	4.3	54.9
British Columbia	2.3	22.0
Nova Scotia	3.6	7.9
Ontario	4.5	32.0

* Robinson P et al., Statistics Canada, released 12 July 2023 [10].

## Data Availability

Not applicable.

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
