# Peer review of "Epidemiology of Chronic Hepatitis C in First Nations Populations in Canadian Prairie Provinces"

_pathogens, 2025, doi:10.3390/pathogens14070693_

Round 1

Reviewer 1 Report

Comments and Suggestions for Authors

Besides, it is a review work

I do not know whether the work should be published in a global medical journal, I suggest a Canadian journal because the subject matter is very specialized

The work is written in great detail, which may discourage the reader from finishing reading

Results in more life years lost than any other infectious disease in Canada - is that true, more than HIV?

The data from 2015 are really out of date with access to DAA drugs since 2015, the use of which is very simple and in some countries does not require a specialist

Regards

Author Response

Results in more life years lost than any other infectious disease in Canada - is that true, more than HIV?

Reply: Yes. This is according to Statistics Canada and other sources including this paper by Kwong JC et al: PLoS One 2012;7(9):e44103.

The data from 2015 are really out of date with access to DAA drugs since 2015, the use of which is very simple and in some countries does not require a specialist

Reply: The reviewer is correct. Some of the data are based on 2015 which is the most recent data available. In Canada except for one province, specialists are still the only physicians who can prescribe DAA.

Reviewer 2 Report

Comments and Suggestions for Authors

This is a comprehensive, well written and well documented review manuscript on the epidemiology of HCV  in First Nations populations in Canadian prairie provinces. The study had to be limited to this geographical area as no reliable data are available for populations living elsewhere. While concentrated on HCV the information is largely relevant for some other infections like HIV. The authors are to be congratulated on the depth of their analysis.

How numerous are  Inuit and Métis populations?

Author Response

This is a comprehensive, well written and well documented review manuscript on the epidemiology of HCV  in First Nations populations in Canadian prairie provinces. The study had to be limited to this geographical area as no reliable data are available for populations living elsewhere. While concentrated on HCV the information is largely relevant for some other infections like HIV. The authors are to be congratulated on the depth of their analysis.

How numerous are  Inuit and Métis populations?

Reply: These data were added (page 1)

Reviewer 3 Report

Comments and Suggestions for Authors

The review by Dunn et al on "Epidemiology of chronic hepatitis C in First Nations populations in Canadian prairie provinces" is a very well written paper, that covers a very import health concern in a very sensitive population. The list of authors appears to be mainly working in Indigenous Services Canada, so they know very well this topic and have worked with First Nations populations. 

Overall, the article requires some modification as I have found some parts very long with too much unnecessary detail for the reader. I understand that the authors want to give the full picture of the situation, but the amount of text should be reduced especially in:

  • Institutional racism in Canadian healthcare
  • Regional definitions of HCV infection: I recommend to move this part to supplementary files, as it disrupts the flow of the article and embed it in too much detailed information (lines 300 to 372)
  • Strategies for change through promising practice

Some questions and remarks:

1- Lines 193 to 197: The idea is not very clear, please reformulate 

2- Hepatitis C is rarely sexually transmitted (incidence less thant 0,07%, Terrault et al, 2013). The current formulation implies that it is also an important transmission route.  Please modify accordingly. 

3- Lines 232 to 252: How do authors explain this arge variation between the different Prairies provinces. 

4- Figure 3 include a table. I suggest to add the figures to the chart.

5- Line 485: The question should be better placed towards the end of the paragraph.

6- Line 535: Implies that viral load is also Point of care. Please separate and modify.

Author Response

Overall, the article requires some modification as I have found some parts very long with too much unnecessary detail for the reader. I understand that the authors want to give the full picture of the situation, but the amount of text should be reduced especially in:

  • Institutional racism in Canadian healthcare

Reply: As per the reviewer’s comment, this part was moved to supplemental appendices

  • Regional definitions of HCV infection: I recommend to move this part to supplementary files, as it disrupts the flow of the article and embed it in too much detailed information (lines 300 to 372)

Reply: as the reviewer’s comment, this part was moved to supplemental appendices.

A brief note to indicate that the 2 sections above have been moved to Supplemental Appendices has been inserted into the text.

Strategies for change through promising practice

Reply: As per the reviewer’s suggestion, we have shortened this section almost 50% down to 600 words from 1066.

Some questions and remarks:

  • Lines 193 to 197: The idea is not very clear, please reformulate 

Reply: as per the reviewer’s comment, we modified the pertinent content (see supplemental appendices).

  • Hepatitis C is rarely sexually transmitted (incidence less than 0,07%, Terrault et al, 2013). The current formulation implies that it is also an important transmission route.  Please modify accordingly. 

Reply: as per the reviewer’s comment, we added “although rare” in the sentence (see supplemental appendices)

  • Lines 232 to 252: How do authors explain this large variation between the different Prairies provinces. 

Reply: The geographical variability in HCV epidemiology likely reflects the large number of often diverse Indigenous cultures and communities with differing levels of socioeconomic status, proximity to larger centres and healthcare access, traditions, healthcare practices, and beliefs. This comment was added to the text.

  • Figure 3 include a table. I suggest to add the figures to the chart.

Reply: The figure and table convey the full message: the figure shows the tendency visually, while the table details the exact numbers. We do not understand exactly what the reviewer suggests for us to do with this, so we have left it as is.

  • Line 485: The question should be better placed towards the end of the paragraph.

Reply: as per the reviewer’s comment, we moved the sentence accordingly.

  • Line 535: Implies that viral load is also Point of care. Please separate and modify.

Reply: as per the reviewer’s comment, we modified the sentence.